# Scalarane Sesterterpenoids Isolated from the Marine Sponge *Hyrtios erectus* and their Cytotoxicity

**DOI:** 10.3390/md20100604

**Published:** 2022-09-25

**Authors:** Huynh Nguyen Khanh Tran, Min Jin Kim, Yeon-Ju Lee

**Affiliations:** 1Marine Natural Products Chemistry Laboratory, Korea Institute of Ocean Science and Technology, 385 Haeyangro, Busan 49111, Korea; 2Department of Marine Biotechnology, University of Science and Technology, 217 Gajungro, Daejeon 34113, Korea

**Keywords:** sponge, *Thorectidae*, *Hyrtios erectus*, sesterterpene, scalarane, γ-hydroxybutenolide, cytotoxicity

## Abstract

Eighteen scalarane sesterterpenoids (**1**–**18**), including eight new derivatives (**1**–**8**), were isolated from the sponge *Hyrtios erectus* (family Thorectidae), the extract of which showed cytotoxicity against the HeLa and MCF-7 cell lines. Of the new derivatives, six compounds (**1**–**6**) were found to contain a γ-hydroxybutenolide moiety capable of reversible stereoinversion at the hydroxylated carbon center. Under the influence of other adjacent functional groups, each derivative exhibited a different stereochemical behavior, which was fully deduced by ROESY experiments. All the isolated compounds were examined for their cytotoxicity by MTS assay using staurosporine as a positive control (IC_50_ 0.18 and 0.13 μΜ against HeLa and MCF-7 cells, respectively), and they were found to show weak growth inhibitory activities against HeLa and MCF-7 cells, with a minimal IC_50_ value of 20.0 μΜ. The compounds containing a γ-hydroxybutenolide moiety (**1**–**3**, **10**, **12**) showed cytotoxicity, with IC_50_ values ranging from 24.3 to 29.9 μΜ, and the most potent derivative was heteronemin (**16**). Although the cytotoxicities of isolated compounds were insufficient to discuss the structure–activity relationship, this research could contribute to expanding the structural diversity of scalaranes and understanding the stereochemical behavior of γ-hydroxybutenolides.

## 1. Introduction

Sesterterpenoids are a group of compounds consisting of a C25 carbon skeleton and are often polycyclic, containing up to five rings. Their biological origins are as diverse as their structures, and they have been isolated from fungi; bacteria; plants; insects; and marine invertebrates, especially sponges [1,2]. The sesterterpenoids isolated from marine sponges have a unique structure that is distinguishable from those isolated from terrestrial organisms; their skeletons consist of four trans-fused cyclohexane rings (A–D), with an optional five-membered heterocycle (E) [3]. This skeleton was named scalarane, as a compound containing this skeleton (scalarin) was first isolated from the marine sponge *Cacospongia scalaris* (a holotype of *Scalarispongia scalaris*) [4,5].

Scalarane sesterterpenoids exhibit a high degree of structural diversity, although their carbon frameworks are rigid. More than 200 scalarane derivatives have been reported, and these can be distinguished by the number of rings (tetra- or pentacyclic), the degree of unsaturation, and the functionalization pattern [1,2,3,4,5]. For tetracyclic scalaranes with an A/B/C/D ring structure, functionalization such as oxygenation at C-12, alkylation at C-19 or C-20, or carbonylation at C-17 or C-18 has been observed. Furthermore, the additional E-ring connected through C-17 and C-18 is often an oxygen-containing heterocycle, with only a few exceptions existing that contain nitrogen (e.g., furan, butyrolactone, pyrrole, or butyrolactam). In addition, the removal or incorporation of the methyl group leads to the formation of nor-, homo-, and bishomoscalaranes.

Scalaranes have been reported to exhibit diverse biological activities, including cytotoxic, anti-microbial, anti-inflammatory, anti-HIV, antitubercular, and antifeedant activities [1,2,3,6,7]. Notably, they have been reported to exhibit significant cytotoxicities against a variety of cancer cell lines, and the pro-apoptotic activity of the scalaranes has been elucidated in several cases. A previous study revealed that 12-*epi*-scalaradial inhibits epidermal growth factor receptor (EGFR)-mediated Akt (protein kinase B) phosphorylation, which may be related to its cytotoxicity against HeLa cells [8]. A recent study showed that a closely related derivative, 12-deacetyl-12-*epi*-scalaradial, triggers apoptosis via the mitogen-activated protein kinase/extracellular signal-regulated kinase (MAPK/ERK) pathway, as well as by activating Nur77 nuclear receptor activity [9]. Another study revealed that two scalarane derivatives induce the apoptotic death of cancer cells via dual inhibition of topoisomerase II and heat shock protein 90 (Hsp90), which are considered to be important molecular targets of anti-cancer medication [10]. Owing to their pharmacological significance, considerable efforts have been made to discover or synthesize scalarane compounds for the discovery of novel anti-cancer therapeutic leads [11,12,13,14,15].

As a part of our ongoing research to discover new chemical entities with potent cytotoxicities, the extracts obtained from marine sponges encountered in field research in various locations were examined for their cytotoxicity. More specifically, it was observed that the extract obtained from the Philippine marine sponge *Hyrtios erectus* exhibited cytotoxicity against HeLa (cervical carcinoma) and MCF-7 (breast cancer) cell lines (51% and 76% growth inhibition at the concentration of 10 μg/mL, respectively). Marine sponges belonging to the genus *Hyrtios* (class: Demospongiae, order: Dictyoceratida, family: Thorectidae) are known as the sources of diverse metabolites, such as indole and β-carboline alkaloids, meroterpenes, sesquiterpenes, and sesterterpenes, as thoroughly reviewed previously [16]. The *Hyrios erectus* sponges have been investigated most extensively, resulting in the isolation of biologically active metabolites such as hyrtiosin derivatives exhibiting cytotoxicity against human epidermoid carcinoma KB cells [17], and hyrtiomazamine exhibiting activity in the B lymphocytes reaction assay [18]. A number of scalarane derivatives have also been isolated from *Hyrtios erectus* sponges; some of the derivatives have shown significant in vitro cytotoxicity against a panel of human cancer cell lines or murine cancer cell lines [19,20,21,22,23].

Intrigued by the cytotoxicity of the extracts obtained from *Hyrtios erectus* described above, efforts have been made to identify the cytotoxic ingredient in this extract, leading to the isolation of eighteen scalaranes, including eight new derivatives. The structure elucidation of new derivatives and the cytotoxicity of isolated scalaranes are reported herein.

## 2. Results

### 2.1. Isolation of Compounds from Hyrtios Erectus

The macerated sponge was freeze-dried and extracted with methanol and dichloromethane, and the combined extract was partitioned between *n*-butanol and water. The *n*-butanol fraction was then partitioned between 15% (*v/v*) aqueous methanol and *n*-hexane. ^1^H nuclear magnetic resonance (NMR) analysis of the obtained fraction indicated that the *n*-hexane fraction contains secondary metabolites, showing signals in the region from *δ*_H_ 6.0 to 7.5 ppm and around 9.5 ppm. The compounds generating these signals could not be specified at this stage due to the overlapping signals from uncharacterized fatty acid derivatives that were predominant in the upfield region from 0 to 2.0 ppm. The *n*-hexane fraction was further fractionated by silica column chromatography, and the obtained fractions were subjected to exhaustive HPLC separation to afford eighteen scalarane compounds (Figure 1, **1**–**18**).

### 2.2. Identification of Isolated Compounds

Among the eighteen isolated compounds, eight (**1**–**8**) were new derivatives, the structures of which have not been previously reported. The remaining ten compounds were identified as previously reported scalaranes based on a comparison of their ^1^H and ^13^C NMR spectra and high-resolution electrospray ionization mass spectrometry (HR-ESIMS) data with literature reports. More specifically, these ten compounds were 12β-acetoxy-16β-methoxy-20α-hydroxy-17-scalaren-19,20-olide (**9**) [16], 12β,20α(β)-dihydroxy-16β-acetoxy-17-scalaren-19,20-olide (**10**) [24], 16-hydroxyscalarolide (**11**) [20], hyrtiolide (**12**) [20], sesterstatin 6 (**13**) [21], 12-*O*-deacetyl-19-deoxyscalarin (**14**) [23], 12-deacetyl-12,18-di-*epi*-scalaradial (**15**) [25], heteronemin (**16**) [26], 12β,16β-diacetoxyscalarolbutenolide (**17**) [27], and 16-*O*-deacetyl-16-*epi*-scalarolbutenolide (**18**) [28] (Figure 1). The structures of the new derivatives were elucidated based on their HR-ESIMS and NMR data, including those obtained in ^1^H-^1^H correlation spectroscopy (COSY), heteronuclear single quantum coherence (HSQC), heteronuclear multiple bond correlation (HMBC), and rotating-frame NOE spectroscopy (ROESY) experiments.

Compound **1** was obtained as an off-white amorphous solid with a molecular formula of C_26_H_40_O_5_, as deduced by HR-ESIMS (*m/z* 433.2936 [M + H]^+^, calculated as 433.2949 for C_26_H_41_O_5_). The IR spectrum of **1** displayed characteristic hydroxyl (3413 and 3300 cm^−1^) and carbonyl (1724 cm^−1^) stretching absorptions, and the UV/Vis spectrum displayed maximum absorptions at 195 and 236 nm, as consistent with other scalarane derivatives reported to date [1,4]. The ^1^H NMR spectrum of **1**, combined with the data mentioned above, suggests the presence of the scalarane carbon framework, with deshielded proton resonances from a hemiacetal moiety (*δ*_H_ 6.06, H-20), two oxymethine moieties (*δ*_H_ 3.98 and 3.61, H-16 and H-12, respectively), and a methoxy group (*δ*_H_ 3.35, 16-OMe) being observed (Table 1). Of the twenty-six carbon resonances present in the ^13^C NMR and DEPT spectra, the resonances that could be assigned to a carbonyl (*δ*_C_ 173.7, C-19), an olefin (*δ*_C_ 158.3 and 140.8, C-17 and C-18, respectively), and a hemiacetal methine (*δ*_C_ 97.7, C-20) revealed the presence of a γ-hydroxybutenolide unit.

The carbon framework of the scalarane was confirmed further by the COSY and HMBC data (Appendix A). In detail, four aliphatic proton spin systems were deduced based on the COSY data. These consisted of three methylenes (C-1 to C-3); one methine and two methylenes (C-5 to C-7); and two sets of one oximethine, one methine, and one methylene in the middle (C-9, C-11, and C-12; C-14 to C-16). Subsequently, these linear systems were assembled using the HMBC correlations with bridgehead singlet methyl protons (H-21/C-3 to C-5; H-22/C-3 to C-5; H-23/C-1, C-5, C-9, and C-10; H-24/C-7 to C-9 and C-14; H-25/C-12 to C-14 and C-18). The absolute stereochemistry assigned by single crystal X-Ray diffraction analyses [29,30,31] or circular dichroism (CD) spectroscopy [32] was taken from the previous literature.

The location of the additional functional groups could also be deduced by the COSY and HMBC data (Appendix A). For one of the oximethines (*δ*_C_ 75.0, C-12), the location was revealed from the COSY correlations between hydrogens attached to C-11 (*δ*_H_ 1.76 and 1.42, H-11) and H-12 (*δ*_H_ 3.51) and the HMBC correlation between the signals H-12 and C-25 (*δ*_C_ 14.7). The other oximethine proton signal (*δ*_H_ 3.98, H-16) showed a COSY correlation with one of the hydrogens attached to C-15 (*δ*_H_ 1.54, H-15β), as well as the HMBC correlations with C-15 (*δ*_C_ 21.7), C-17 (*δ*_C_ 158.3), and C-18 (*δ*_C_ 140.8), indicating the C-16 position. As the proton signal from the methoxy group showed an HMBC correlation with C-16 (*δ*_C_ 67.7), it was deduced that the hydroxyl group was attached to C-12 and the methoxy group to C-16 (Figure 2). In addition, the diaxial coupling (*J* = 10.8 Hz) of the signal corresponding to H-12 (δ_H_ 3.61) and H-11 (δ_H_ 1.76) suggested a β-orientation for the hydroxyl group. The β-pseudoequatorial orientation of H-16 was deduced by the small coupling constant (*J* = 4.2 Hz) of the corresponding signal at δ_H_ 3.98 [33,34]; according to previous literature, a pseudoaxial α-hydrogen would exhibit a coupling constant larger than 9.0 Hz [8,22]. The carbonyl group of the γ-hydroxybutenolide moiety was attached to C-18 and the hemiacetal to the C-17, as the HMBC correlations with the carbon signals corresponding to those moieties (C-17, C-19, and C-20) indicated (H-16/C-17; H-25/C-18; H-20/C-17 to C-19).

The overall structure of compound **1** was very similar to that of the compound reported by Jeon et al. (Figure 3, **19**), with the exception that the acetoxy group was substituted with a hydroxyl group at C-12 of compound **1** [33]. Despite this subtle structural modification of the structure, the chemical behaviors of the two compounds differed dramatically given the interconversion of the stereochemistry at C-20. It has been reported previously that γ-hydroxybutenolide tends to possess a labile stereogenic center at the α position (Figure 3) [35,36,37,38]. Owing to the interconversion of stereochemistry through the ring-opening/ring-closing process, diverse natural products that contain the γ-hydroxybutenolide moiety tend to exist as the mixtures of epimers [39]. Although the previously reported 12-*O*-acetylated derivative of **1** (**19**) was also detected as a mixture of C-20 epimers in a ratio of 1.0:2.5, as observed by ^1^H NMR experiments in CDCl_3_, only a single epimer was observed in the ^1^H and ^13^C NMR spectra of **1** obtained in CDCl_3_. It should also be noted that many broad signals were observed in both the ^1^H and ^13^C NMR spectra of **19**, which hindered the subsequent NMR-based structural determination [33]. This problem has been mitigated by changing the solvent to pyridine-*d*_5_; the signals have been significantly enhanced, along with the changes in the ratio of epimers to 1.0:10.1. In contrast, the NMR signals for compound **1** obtained in CDCl_3_ were sufficiently clear to allow their full assignment (Appendix A). The relative stereochemistry at the C-20 position of the detected diastereomer (**1**) could be assigned based on the ROESY correlation between H-20 (*δ*_H_ 6.06) and H-16 (*δ*_H_ 3.98) (Figure 2).

Compound **2**, isolated as an off-white amorphous solid, exhibited a pseudomolecular ion at *m/z* 433.2928 ([M + H]^+^, calculated as 433.2949 for C_26_H_41_O_5_), which suggested the same molecular formula as compound **1**. The ^1^H and ^13^C NMR spectra were comparable to those of compound **1** (Table 1), implying that both compounds shared an identical carbon skeleton. However, upon a detailed comparison of the NMR data of compounds **1** and **2**, a significant difference was observed in the splitting pattern of the signal corresponding to H-16 (*δ*_H_ 4.10, dd, *J* = 10.2, 7.2 Hz for **2**), suggesting the pseudoaxial α-orientation of H-16 in compound **2**. The ROESY correlation between the signals at *δ*_H_ 3.47 (s, 16-OCH_3_) and 6.15 (s, H-20) verified that the stereochemistry at C-20 remained unchanged from that of **1** (Figure 2); thus, compound **2** was identified as the C-16 epimer of compound **1**.

Compound **3** was obtained as an off-white amorphous solid, and its molecular formula was deduced as C_25_H_38_O_5_ based on the observation of its sodium adduct ion at *m/z* 441.2604 (calculated as 441.2611 for C_25_H_38_O_5_Na) in the HR-ESIMS data. The ^1^H and ^13^C NMR spectra of **3** showed a similar pattern to those of **2**; namely deshielded proton resonances from a hemiacetal moiety (*δ*_H_ 6.96, H-20) and two oxymethines (*δ*_H_ 5.12 and 3.65, H-16 and H-12, respectively), and a large coupling constant for the H-16 signal (*J* = 10.2 Hz) in the ^1^H NMR spectrum (Table 1). In addition, ^13^C NMR signals originating from an α, β-unsaturated carbonyl (*δ*_C_ 165.3, 139.2, and 174.8, C-17−19, respectively), a hemiacetal (*δ*_C_ 99.4, C-20), and two oxymethines (*δ*_C_ 76.2 and 66.4, C-12 and C-16, respectively) were observed. The critical difference was the absence of oxymethyl signals in the ^1^H and ^13^C NMR spectra of **3** compared to that of **2** (*δ*_H_ 6.96, *δ*_C_ 76.2, 16-OMe). This observation is consistent with high-resolution mass data, suggesting the loss of the methyl group in **2**. Based on these observations, the structure of **3** was expected to be a 16-*O*-demethylated derivative of **2**, with the exception of the stereochemistry at C-20. The ROESY correlation between H-16 (*δ*_H_ 5.12) and H-20 (*δ*_H_ 6.96) indicates the β-orientation of 20-OH (Figure 2).

An additional pair of C-16 epimers at the γ-hydroxybutenolide moiety was isolated individually, namely compounds **4** and **5**, which are regioisomers of compounds **1** and **2**, respectively. In the case of compounds **4** and **5**, an HMBC correlation from H-25 was observed with C-19 (*δ*_C_ 170.7 for **4** and 169.1 for **5**), which could be assigned as the β-carbon of an α, β-unsaturated carbonyl moiety based on its chemical shift, whereas an HMBC cross peak was found for H-25 and an α-carbon (*δ*_C_ 140.8 for **1** and 140.4 for **2**) in the case of compounds **1** and **2** (Table 2, Figure 2).

A more critical difference in the ^1^H NMR spectra between these regioisomers (i.e., compounds **1** and **2** versus compounds **4** and **5**) is that compounds **1** and **2** exist as a single diastereomer, while compounds **4** and **5** are mixtures of C-19 epimers (3:1 and 2:1, respectively), even when the spectra were recorded in pyridine-*d*_5_. It should be noted that hyrtiolide (**12**) and sesterstatin 6 (**13**), the 16-*O*-demethylated derivatives of **4** and **5**, were observed as a single diastereomer under NMR analysis in CD_3_OD and CDCl_3_:DMSO-*d*_6_(5:1), respectively, as reported previously [20,21]. The major epimers of **4** and **5** were those bearing an α-hydroxyl group at C-19, as determined by the ROESY correlations between H-19 and H-25 (Figure 2). The stereochemistry at C-12 and C-16 was elucidated in the same way as for compounds **1** and **2**. The β-orientations of the hydroxyl groups at C-12 of **4** and **5** were deduced by diaxial coupling between H-12 and H-11β (*J* = 10.8 and 11.4 Hz, respectively), as well as a ROESY correlation between H-12 and H-9. The H-16 of compound **4** was assigned to have a pseudoequatorial β-orientation (*δ*_H_ 4.00, d, *J* = 3.6 Hz), while that of **5** was assigned as an α-pseudoaxial hydrogen (*δ*_H_ 4.09, dd, *J* = 9.3, 7.5 Hz) based on the *J* value of each signal originating from those.

In the ^1^H NMR data of compound **6**, characteristic signals were observed for a 1,1-disubstituted *cis*-olefin (*δ*_H_ 6.11 and 6.57, H-15 and H-16, respectively), while the signals originating from a methoxymethine (*δ*_H_ 4.09, H-16) and methoxy (*δ*_H_ 3.54, 16-OMe) group observed for compound **5** were not found (Table 3). These observations are in accordance with the high-resolution mass data of **6**, which showed the presence of a pseudomolecular ion at *m*/*z* 401.2695 ([M + H]^+^, calculated as 401.2686 for C_25_H_37_O_4_), suggesting the loss of methanol from **5**. The COSY correlations between the olefinic signals (*δ*_H_ 6.11 and 6.57, H-15 and H-16, respectively) and a methine signal at *δ*_H_ 2.19 (H-14) and the HMBC correlation between one of the olefinic hydrogen (*δ*_H_ 6.57, H-16) and the β-carbon (*δ*_C_ 167.3, C-18) of the α, β-unsaturated carbonyl allowed the conjugated diene to be placed in the D ring (Appendix A). The signal at *δ*_H_ 4.57 (H-12) was assigned to the axial oxymethine (*J* = 11.4, 4.2 Hz), which indicated that the 12-OH group adopted the β-orientation. Moreover, a ROESY correlation between the 25-methyl signal (*δ*_H_ 1.26) and the hemiacetal hydrogen resonance (*δ*_H_ 7.02, H-19) indicated that 19-OH was in the α-orientation (Figure 2).

Based on the corresponding ^13^C NMR data and HR-ESIMS results, the molecular formula of compound **7** was deduced as C_25_H_38_O_4_ (*m/z* 425.2649, calculated as 425.2662 for C_25_H_38_O_4_Na) and ^13^C NMR data. The ^1^H and ^13^C NMR spectra of **7** closely resembled those of 12-deacetyl-12, 18-di-*epi*-scalaradial (**15**) (Table 3), which was first isolated from *Spongia idia* [25]; however, the disappearance of the ^1^H NMR signal corresponding to an aldehyde moiety along with the upfield shift of the ^13^C NMR signal from the carbonyl group indicated that one of the aldehyde groups had been substituted with a carboxylate moiety (*δ*_C_ 176.7, C-19). The location of this carboxylate group was deduced by the correlations between the methyl signal at *δ*_H_ 1.13 (3H, s, H-25) and the methine signal at *δ*_C_ 52.1 (C-18), in addition to an additional correlation between the methine signal at *δ*_H_ 4.27 (1H, s, H-18) and the carboxylate signal at *δ*_C_ 176.6 (C-19) in the HMBC spectrum (Appendix A). The β-orientation of H-18, which was consistent with the stereochemistry of compound **15**, was revealed by the ROESY data (Figure 2), with a correlation being observed between the methyl signal at *δ*_H_ 1.13 (3H, s, H-25) and the methine signal at *δ*_H_ 4.27 (1H, s, H-18).

Compound **8** was expected to be a norsesterterpenoid, based on the pseudomolecular ion peak observed in its HR-ESIMS spectrum (*m/z* 413.2667 [M+Na]^+^, calculated as 413.2662 for C_24_H_38_O_4_Na) and the number of signals that appeared in its ^13^C NMR spectrum. The characteristic signals of two oxymethines at *δ*_H_ 3.74 (H-12)/*δ*_C_ 77.0 (C-12) and *δ*_H_ 5.13 (H-16)/*δ*_C_ 69.4 (C-16) and an olefin at *δ*_H_ 7.94 (H-18)/*δ*_C_ 149.7 (C-18) and *δ*_C_ 133.0 (C-17) were observed in the ^1^H and ^13^C NMR spectra of **8** (Table 3). The HMBC correlation between the 25-methyl signal (*δ*_H_ 1.40) and one of the oxymethine (*δ*_C_ 77.0) suggested hydroxylation at C-12 (Appendix A). Based on the HMBC correlations from H-12 to C-24 (*δ*_C_ 16.9) and C-18, and H-18 to the other oxymethine C-16, the spin system could be extended further to C-16 through C-17. The ^13^C NMR signal at *δ*_C_ 172.0 (C-19), combined with high-resolution mass data, indicates the presence of carboxylate. The location of this carboxylate was confirmed by the presence of an α, β-unsaturated carbonyl functionality, as deduced from the downfield shift of the signals of H-18 and C-18, as well as the HMBC correlation from H-18 to C-19. Based on these observations, it was suspected that the structure of **8** was similar to that of 12-*O*-deacetylnorscalaral B, which had been previously isolated from *Hyatella intestinalis* [34]. However, these two compounds differed in their degree of oxidation at C-17, with **8** containing a carboxylate group, and 12-*O*-deacetylnorscalaral B possessing an aldehyde moiety. The relative stereochemistry at C-12 and C-16 was also contrasting; in the case of 12-*O*-deacetylnorscalaral B, both 12- and 16-OH adopt the α-orientation, whereas those of compound **8** adopt the β-orientation. The stereochemistry at each stereogenic center was further confirmed by the relatively large coupling constants of H-12 (*J* = 10.8 Hz) and H-16 (*J* = 8.7 Hz), which are indicatives of axial and pseudoaxial positions, respectively.

### 2.3. Cytotoxicity of Isolated Scalarane Derivatives

To identify the biologically active compound present in the extract, all isolated compounds were subjected to cytotoxicity assessments against HeLa and MCF-7 cancer cell lines (Table 4). With the exception of compound **5**, all compounds showed weak cytotoxicity, with higher potency against MCF-7 cells than against HeLa cells. In addition, compounds **7** and **8** resulted in the viability of both cancer cell lines over 70% at a concentration of 80 μΜ, thereby indicating that the additional five-membered ring is crucial for cytotoxicity. Furthermore, the compounds bearing a γ-hydroxybutenolide moiety (**1**−**3**, **10**, and **12**) showed cytotoxicity against the MCF-7 cells (IC_50_ < 30.0 μΜ), especially in the case of those bearing a carbonyl group at C-19 and a hydroxyl group at C-12 (**1**−**3** and **10**). Heteronemin (**16**) was found to exhibit the highest growth inhibitory activity against the MCF-7 cell line, with an IC_50_ value of 20.0 μΜ; it should be noted that the previously reported IC_50_ values of heteronemin against the same cell line were 0.29 and 0.70 μΜ [40,41]. The cause of this significant difference in activities is currently unknown; however, repeated experiments provided consistent results for compound **16**. Moreover, the result obtained for compound **9**, namely an IC_50_ value of 43.8 μΜ, matches with that previously reported (IC_50_ = 40.3 μΜ) [22], which implies the consistency and reliability of our cytotoxicity data.

## 3. Discussion

This study was initiated due to the promising results obtained during the screening of the *Hyrtios erectus* extract for its cytotoxicity against human cancer cell lines (HeLa and MCF-7 cells, 51 and 76% growth inhibition at 10 μg/mL, respectively), as described in the introduction. The fractionation of the extract was followed by ^1^H NMR analysis, which provided the rationale for selecting fractions for further analysis. More specifically, the *n*-hexane fraction obtained by solvent partitioning between *n*-butanol and water followed by an additional partition between 15% (*v/v*) aqueous methanol and *n*-hexane showed several characteristic signals. Further fractionation and separation provided eighteen scalarane derivatives (**1**−**18**), the structures of which were elucidated by ^1^H and ^13^C NMR spectroscopy and high-resolution mass spectrometry. In addition to these 1D NMR data, 2D NMR data (including HSQC, HMBC, COSY, and ROESY spectra) were obtained and interpreted for the eight new derivatives.

Among the isolated compounds, ten (**1**−**6**, **9**, **10**, **12**, and **13**) were those bearing a γ-hydroxybutenolide moiety, which is capable of reversible stereoinversion. More specifically, the majority of γ-hydroxybutenolide derivatives bearing a carbonyl group at the C-19 position (**1–3** and **9**) were detected as single epimers at the C-20 position when their NMR spectra were recorded in CDCl_3_ or pyridine-*d*_5_; one exception was compound **10**, which contained an acetoxy group at the C-12 position. In the case of regioisomers bearing a carbonyl group at C-20 (**4**, **5**, **12**, and **13**), the compounds containing a methoxy group at C-16 (**4** and **5**) were detected as a mixture of C-19 epimers, whereas those containing a 16-hydroxy group (**12** and **13**) were detected as single epimers, regardless of the stereochemistry at C-16. The stereochemistry of each new derivative could be deduced by ROESY experiments as the structures present under the conditions of NMR analysis. Parallel NMR experiments performed under the same solvent condition are needed to confirm the effect of adjacent functional groups on the stereochemistry of γ-hydroxybutenolide. However, it was difficult to compare the data under the same conditions, as NMR solvent selection for natural products mainly depends on the solubility of each compound. Thus, to carry out strictly controlled experiments, specially designed and chemically synthesized γ-hydroxybutenolides are required.

The biosynthetic pathways and the involved gene clusters for scalaranes have not been sufficiently discovered to date, as indicated by a limited number of previous reports. However, it was experimentally demonstrated that squalene-hopene cyclase (SHC) could transform geranylfarnesol or its phosphate ester into scalarenol (**20**) through the cationic cyclization cascade (Figure 4) [42,43]. Compound **20**, bearing a 6/6/6/6 tetracyclic scalarane skeleton, might be a common intermediate in the biosynthesis of diverse scalaranes. The compounds bearing a lactone (**14**), a γ-hydroxybutenolide (**1**–**6**, **9**–**13**), and a tetracyclic skeleton (**7**, **8**, **15**) isolated in this study can be synthesized by the simple biochemical transformations of **20**, such as oxidation and lactonization. SHC has been found in bacteria mostly, but also in fungi, cyanobacteria, and plants [44,45,46,47]; it is highly probable that scalaranes might be synthesized by the (micro)organisms symbiotic or associated with the sponges.

Considering the initial results obtained using the crude extract, the cytotoxicities recorded for the isolated compounds were disappointing. All of the obtained fractions were investigated thoroughly in order to identify the compounds responsible for the cytotoxicity of the extract. The polar aqueous methanol fraction obtained after the final solvent partition was also subjected to further separation, only to figure out that this fraction mostly contained primary metabolites such as nucleosides, amino acids, and short peptides. Since the isolated compounds showed only weak cytotoxicities, our results were insufficient to discuss the structure–activity relationship. However, this research would contribute to expanding the structural diversity of scalaranes and understanding the stereochemical behavior of γ-hydroxybutenolides.

## 4. Materials and Methods

### 4.1. General Experimental Procedures

The 1D and 2D NMR spectra were recorded using an ASCEND 600 spectrometer (Bruker BioSpin Gmbh, Rheinstetten, Germany) at 298 K. All chemical shifts are reported in ppm from tetramethylsilane, using solvent resonances resulting from incomplete deuteration as the internal reference. The HR-ESIMS data were obtained using a TripleTOF 5600+ system (SCIEX, Framingham, MA, USA). Specific optical rotations were measured using an Autopol III S2 Polarimeter (Rudolph Research Analytical, Hackettstown, NJ, USA). IR spectra were recorded using an FT/IR-4100 spectrometer (JASCO Inc., Easton, MD, USA), while the UV–visible spectra were measured using a Shimadzu UV-1650PC spectrophotometer. Semi-preparative HPLC was performed using a Waters^TM^ 1525 binary pump and a UV/Visible 2489 detector (Tepnel Pharma Services Limited, Livingston, Scotland). HPLC was performed using a YMC-Pack Pro C-18 column (250 × 10 mm l.D., S-5 µm, 12 nm). Silica gel (0.04–0.063 mm particle size, Merck) and RP-18 (0.04–0.063 mm particle size, Merck) were used for carrying out flash column chromatography. Thin layer chromatography (TLC) was performed using Merck silica gel 60 F_254_ and RP-18 F_254_ plates. All reagents were purchased from Sigma-Aldrich (Merck KGaA, Darmstadt, Germany).

### 4.2. Sponge Material

The sponge, *Hyrtios erectus*, was collected from the coral reef of Bohol island in the Philippines (9°44′46.93′′ N, 124°35′10.82′′ E) in March of 2016, at a depth of 15 m by hand using SCUBA. The general depiction of the sponge surface resembles that reported by Shin et al. [48], while the gross morphological features were similar to those of *Hyrtios erectus* and *Heteronema erecta* (Keller, 1889) [49]. Voucher specimens were deposited at the Korea Institute of Ocean Science and Technology (registry no. 163PIL-483), Busan, Korea.

### 4.3. Extraction and Isolation

The wet sponge (1021.0 g in total) was macerated to obtain specimens measuring approximately 1.0–1.5 cm, which were dried under the fume hood at ambient temperature (resulting mass 728.0 g) and then extracted using methanol (300 mL × 2) and dichloromethane (300 mL × 1) at room temperature. The combined extract was partitioned between *n*-butanol (10.7 g) and water (36.0 g), and the *n*-butanol fraction was further partitioned between 15% aqueous methanol and *n*-hexane. Guided by the ^1^H NMR spectra, the *n*-hexane fraction (5.4 g) was subjected to a normal phase column chromatography with a stepped gradient *n*-hexane/ethyl acetate solvent to yield thirteen fractions. The sixth to tenth fractions (75 to 90% (*v*/*v*) ethyl acetate in *n*-hexane) were separated further by size-exclusion chromatography (Sephadex LH-20, 20% aqueous MeOH) and reversed-phase HPLC using aqueous methanol as the mobile phase and a UV detector (*λ* = 210 and 230 nm) to afford compounds **1** (10.6 mg), **2** (6.1 mg), **3** (1.9 mg), **4** (3.8 mg), **5** (1.7 mg), **6** (3.1 mg), **7** (1.2 mg), **8** (2.1 mg), **9** (1.8 mg), **10** (52.3 mg), **11** (7.7 mg), **12** (4.5 mg), **13** (17.1 mg), **14** (4.1 mg), **15** (6.2 mg), **16** (3.7 mg), **17** (1.6 mg), and **18** (6.8 mg).

12β,20β-Dihydroxy-16α-methoxy-17-scalaren-19,20-olide *(**1**)*. White amorphous powder; [*α*]_D_^23^ + 100.0 (*c* 0.5, methylene chloride (MC)); UV *λ*_max_ (MC) (log *ε*) 195 (0.67) and 236 (0.76) nm; IR (ATR) *ν*_max_: 3413, 3300, 2925, 2858, 1724, 1455, and 1017 cm^−1^; ^1^H and ^13^C NMR data in CDCl_3_, see Table 1; HR-ESIMS *m/z*: 433.2936 [M + H]^+^ (calcd. for C_26_H_41_O_5_, 433.2949, *Δ* −2.9 ppm).

12β,20α-Dihydroxy-16β-methoxy-17-scalaren-19,20-olide *(**2**)*. Off-white amorphous powder; [*α*]_D_^22^ + 10.0 (*c* 0.5, MC); UV *λ*_max_ (MC) (log *ε*) 235 (3.62) nm; IR (ATR) *ν*_max_: 3297, 2925, 2858, 1724, 1466, and 1021 cm^−1^; ^1^H and ^13^C NMR data in CDCl_3_, see Table 1; HR-ESIMS *m/z*: 433.2928 [M + H]^+^ (calcd. for C_26_H_41_O_5_, 433.2949, *Δ* −4.7 ppm).

12β,16β,20β-Trihydroxy-17-scalaren-19,20-olide *(**3**)*. Off-white amorphous powder; [*α*]_D_^23^ + 100.0 (*c* 0.4, MC); UV *λ*_max_ (MC) (log *ε*) 239 (0.56) nm; IR (ATR) *ν*_max_: 3406, 3218, 2918, 2858, 1721, 1452, 1374, and 1021 cm^−1^; ^1^H and ^13^C NMR data in C_5_D_5_N, see Table 1; HR-ESIMS *m/z*: 441.2604 [M + Na]^+^ (calcd. for C_25_H_38_O_5_Na, 441.2611, *Δ* −1.7 ppm).

12β,19α(β)-Dihydroxy-16α-methoxy-17-scalaren-19,20-olide *(**4**)*. Off-white amorphous powder; [*α*]_D_^24^ − 124.2 (*c* 0.5, MC); UV *λ*_max_ (MC) (log *ε*) 231 (2.32) nm; IR (ATR) *ν*_max_: 3385, 2911, 2861, 1749, 1458, 1371, and 1021 cm^−1^; ^1^H and ^13^C NMR data in CDCl_3_, see Table 2; HR-ESIMS *m/z*: 433.2961 [M + H]^+^ (calcd. for C_26_H_41_O_5_, 433.2949, *Δ* +2.9 ppm).

12β,19α(β)-Dihydroxy-16β-methoxy-17-scalaren-19,20-olide *(**5**)*. Off-white amorphous powder; [*α*]_D_^24^ + 9.0 (*c* 0.4, MC); UV *λ*_max_ (MC) (log *ε*) 238 (0.70) nm; IR (ATR) *ν*_max_: 3356, 2922, 2861, 1745, 1452, 1377, 1265, and 1021 cm^−1^; ^1^H and ^13^C NMR data in CDCl_3_, see Table 2; HR-ESIMS *m/z*: 433.2961 [M + H]^+^ (calcd. for C_26_H_41_O_5_, 433.2949, *Δ* +2.9 ppm).

12β,19α-Dihydroxy-14,15-dehydrate-17-scalaren-19,20-olide *(**6**)*. Off-white amorphous powder; [*α*]_D_^25^ − 139.4 (*c* 0.5, MC); UV *λ*_max_ (MC) (log *ε*) 249 (3.48) nm; IR (ATR) *ν*_max_: 3374, 2922, 2858, 1745, 1458, 1377, and 1028 cm^−1^; ^1^H and ^13^C NMR data in C_5_D_5_N, see Table 3; HR-ESIMS *m/z*: 401.2695 [M + H]^+^ (calcd. for C_25_H_37_O_4_, 401.2686, *Δ* +2.2 ppm).

12-Deacetyl-18-*epi*-carboxylic-12-*epi*-scalaral *(**7**)*. Off-white amorphous powder; [*α*]_D_^25^ − 6.1 (*c* 0.2, MC); UV *λ*_max_ (MC) (log *ε*) 238 (0.53) nm; IR (ATR) *ν*_max_: 3413, 2925, 2858, 1710, 1458, 1381, 1261, 1017, and 734 cm^−1^; ^1^H and ^13^C NMR data in C_5_D_5_N, see Table 3; HR-ESIMS *m/z*: 425.2649 [M + Na]^+^ (calcd. for C_25_H_38_O_4_Na, 425.2662, *Δ* −3.1 ppm).

2-*O*-Deacetyl-12,16-di*-epi*-norscalaral B *(**8**)*. Off-white amorphous powder; [*α*]_D_^25^ − 27.3 (*c* 0.3, MC); UV *λ*_max_ (MC) (log *ε*) 231 (2.37) nm; IR (ATR) *ν*_max_: 3470, 3127, 2925, 2855, 1717, 1458, 1377, 1303, 1265, 1027, and 735 cm^−1^; ^1^H and ^13^C NMR data in C_5_D_5_N, see Table 3; HR-ESIMS *m/z*: 413.2667 [M + Na]^+^ (calcd. for C_24_H_38_O_4_Na, 413.2662, *Δ* +1.1 ppm).

### 4.4. Biological Assays

The MTS assay was performed using the CellTiter 96^®^ AQ_ueous_ One Solution Cell Proliferation Assay (Promega, Madison, WI, USA). For this purpose, HeLa and MCF-7 cells were plated in 384-well clear plates at densities of 3700 cells/well and 2000 cells/well, respectively. The seeded cells were incubated for 24 h at 37 °C in 5% (*v*/*v*) CO2 and then treated with the compounds at seven different final concentrations (1.25, 2.5, 5.0, 10, 20, 40, and 80 μΜ) for 48 h. DMSO was used as the vehicle control, and staurosporine was used as a positive control. The viable cell numbers were determined based on the concentration of formazan resulting from tetrazolium conversion, and the absorbance was measured at 490 nm using an EnVision Xcite Multilabel Reader (PerkinElmer, Waltham, MA, USA).

## Figures and Tables

**Figure 1 marinedrugs-20-00604-f001:**
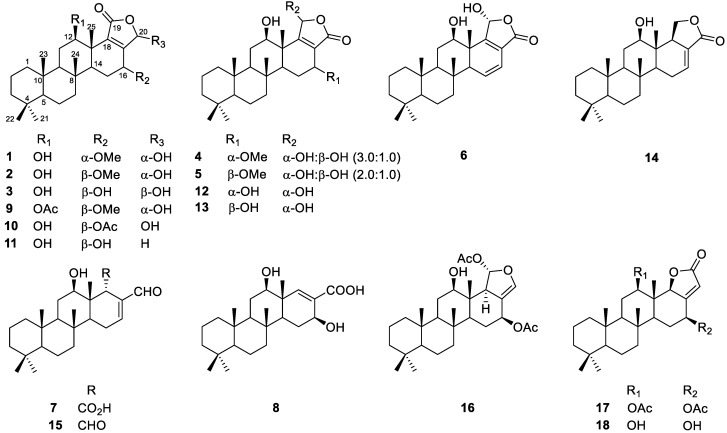
Structures of isolated compounds **1**−**18**.

**Figure 2 marinedrugs-20-00604-f002:**
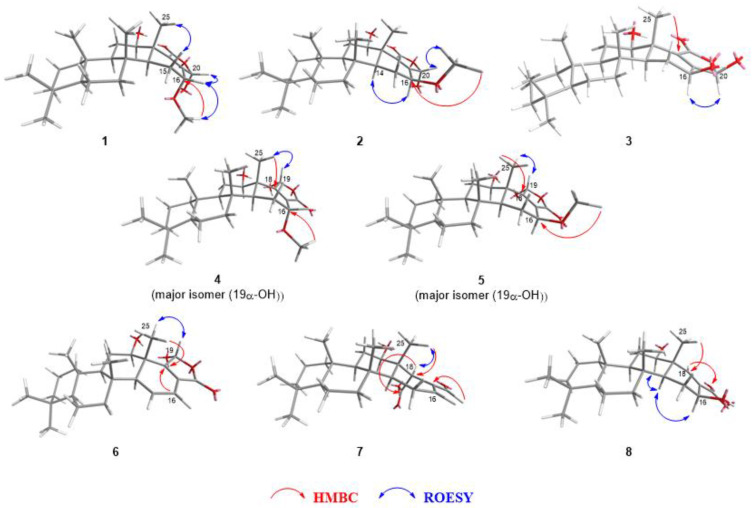
Key HMBC and ROESY correlations for compounds **1**−**8**.

**Figure 3 marinedrugs-20-00604-f003:**
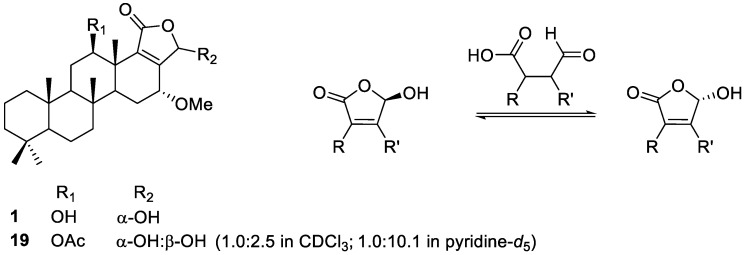
Structures of the γ-hydroxybutenolide derivatives (**1**, **19**) and their stereoisomerization.

**Figure 4 marinedrugs-20-00604-f004:**
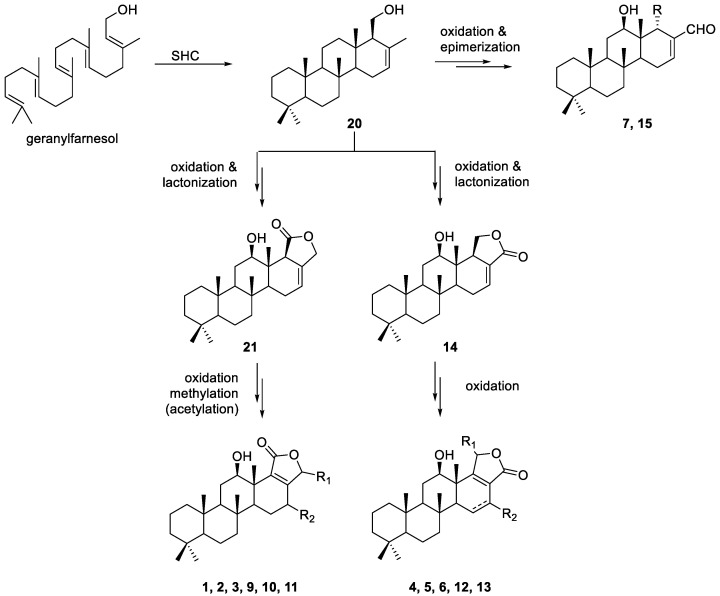
The plausible biosynthetic pathway for isolated scalarane derivatives.

**Table 1 marinedrugs-20-00604-t001:** ^13^C (150 MHz) and ^1^H (600 MHz) NMR data for compounds **1**−**3**.

Position	1 ^a^		2 ^a^		3 ^b^	
	*δ*_C_^c^, Type	*δ*_H_ (*J* in Hz)	*δ*_C_^c^, Type	*δ*_H_ (*J* in Hz)	*δ*_C_^c^, Type	*δ*_H_ (*J* in Hz)
1	39.7, CH_2_	1.65, m	40.0, CH_2_	1.72, m	40.3, CH_2_	1.61, d (13.2)
		0.70, m ^d^		0.77, m ^d^		0.73, m
2	18.2, CH_2_	1.53, m ^d^	18.7, CH_2_	1.59, m ^d^	19.3, CH_2_	1.54, m
		1.36, m ^d^		1.42, m ^d^		1.35, m ^d^
3	42.1, CH_2_	1.30, m	42.3, CH_2_	1.38, d (13.8)	42.8, CH_2_	1.33, m ^d^
		1.06, m		1.09, m ^d^		1.13, dt (13.8, 4.2)
4	33.3, C		33.4, C		33.8, C	
5	56.6, CH	0.70, m ^d^	56.9, CH	0.77, m ^d^	57.1, CH	0.71, m
6	18.6, CH_2_	1.53, m ^d^	18.3, CH_2_	1.59, m ^d^	18.9, CH_2_	1.44, m
		1.36, m ^d^		1.42, m ^d^		1.28, m
7	42.7, CH_2_	1.76, m ^d^	41.8, CH_2_	1.73, m	41.9, CH_2_	1.68, d (12.6)
		0.86, m		0.91, m		0.63, m ^d^
8	36.8, C		37.6, C		37.6, C	
9	57.8, CH	0.84, m ^d^	58.1, CH	0.85, m	58.2, CH	0.65, d (12.6)
10	37.4, C		37.3, C		37.9, C	
11	25.5, CH_2_	1.76, m ^d^	25.6, CH_2_	1.83, m	26.6, CH_2_	1.96, m ^d^
		1.42, m		1.49, m		1.62, d (10.8)
12	75.0, CH	3.61, dd (10.8, 4.2)	75.2, CH	3.60, dd (10.8, 4.2)	76.2, CH	3.65, dd (10.8, 4.2)
13	41.4, C		42.8, C		43.7, C	
14	49.7, CH	1.22, m	54.0, CH	1.09, m^d^	54.9, CH	1.00, d (12.0)
15	21.7, CH_2_	2.06, d (14.4)	23.6, CH_2_	2.30, dd (12.0, 7.2)	28.8, CH_2_	2.39, dd (12.6, 7.2)
		1.54, m		1.53, m		1.99, m ^d^
16	69.7, CH	3.98, d (4.2)	74.5, CH	4.10, dd (10.2, 7.2)	66.4, CH	5.12, dd (10.2, 7.2)
17	158.3, C		160.0, C		165.3, C	
18	140.8, C		140.0, C		139.2, C	
19	173.7, C		172.7, C		174.8, C	
20	97.7, CH	6.06, s	96.9, CH	6.15, s	99.4, CH	6.96, s
21	33.3, CH_3_	0.78, s	33.5, CH_3_	0.84, s ^d^	33.9, CH_3_	0.87, s
22	21.3, CH_3_	0.75, s	21.4, CH_3_	0.81, s	21.8, CH_3_	0.80, s
23	15.9, CH_3_	0.78, s	16.1, CH_3_	0.84, s^d^	16.5, CH_3_	0.77, s
24	17.2, CH_3_	0.83, s	17.5, CH_3_	0.90, s	17.8, CH_3_	0.85, s
25	14.7, CH_3_	1.02, s	16.5, CH_3_	1.15, s	17.2, CH_3_	1.34, s
16-OMe	57.4, CH_3_	3.35, s	57.7, CH_3_	3.47, s		

^a^ Data obtained in CDCl_3_. ^b^ Data obtained in pyridine-*d*_5_. ^c^ Carbon atoms correlating with the corresponding protons in the HSQC spectrum. ^d^ Overlapped with other signals.

**Table 2 marinedrugs-20-00604-t002:** ^13^C (150 MHz) and ^1^H (600 MHz) NMR data for compounds **4** and **5**
^a^.

Position	4		5	
	*δ*_C_^b^, Type	*δ*_H_ (*J* in Hz)	δ_C_ ^b^, Type	*δ*_H_ (*J* in Hz)
1	39.9, CH_2_	1.60, m ^c^	40.0, CH_2_	1.68, m
		1.43, m ^c^		0.78, m ^c^
2	18.3/18.7 ^d^, CH_2_	1.60, m ^c^	18.7, CH_2_	1.57, m ^c^
		1.43, m ^c^		1.42, m ^c^
3	42.2, CH_2_	1.37, m	42.1, CH_2_	1.37, m
		1.15, m		1.13, m
4	33.4, C		33.3, C	
5	56.6, CH	0.80, m ^c^	56.7, CH	0.78, m ^c^
6	18.7/18.3 ^d^, CH_2_	1.68, d (12.6)	18.3, CH_2_	1.57, m ^c^
		0.80, m ^c^		1.42, m ^c^
7	41.5, CH_2_	1.79, d (12.6)	41.5, CH_2_	1.82, m
		0.94, d (12.6) ^c^		0.90, m
8	37.6, C		37.5, C	
9	58.7, CH	0.94, d (12.6) ^c^	58.7, CH	0.87, m ^c^
10	37.0, C		37.6, C	
11	25.8, CH_2_	1.82, m	26.0, CH_2_	1.79, m
		1.55, d (12.6)		1.52, m
12	74.1, CH	3.80, dd (10.8, 4.5)	74.4, CH	3.80, dd (11.4, 4.2)
13	45.3, C		44.7, C	
14	49.0, CH	1.44, m	53.4, CH	1.42, m^c^
15	22.5, CH_2_	2.06, d (15.0)	24.6, CH_2_	2.25, m
		1.50, dd (13.5, 3.9)		1.63, m
16	69.2, CH	4.00, d (3.6)	73.8, CH	4.09, dd (9.3, 7.5)
17	126.9, C		128.6, C	
18	170.7, C		169.1, C	
19	95.9, CH	6.12, d (1.2)	95.4, CH	6.18, s
20	170.4, C		169.2, C	
21	33.4, CH_3_	0.81, s	33.4, CH_3_	0.84, s ^c^
22	21.4, CH_3_	0.85, s ^c^	21.4, CH_3_	0.81, s
23	16.3, CH_3_	0.85, s ^c^	16.7, CH_3_	0.84, s ^c^
24	17.9, CH_3_	0.90, s	17.6, CH_3_	0.91, s
25	15.4, CH_3_	1.11, s	16.6, CH_3_	1.21, s
16-OMe	57.7, CH_3_	3.46, s	58.0, CH_3_	3.54, s

^a^ Data obtained in CDCl_3_. ^b^ Carbons correlating with the corresponding protons in the HSQC spectrum. ^c^ Overlapped with other signals. ^d^ Overlapped with other signals.

**Table 3 marinedrugs-20-00604-t003:** ^13^C (150 MHz) and ^1^H (600 MHz) NMR data for compounds **6**−**8**^a^.

Position	6		7		8	
	*δ*_C_^b^, Type	*δ*_H_ (*J* in Hz)	*δ*_C_^b^, Type	*δ*_H_ (*J* in Hz)	*δ*_C_^b^, Type	*δ*_H_ (*J* in Hz)
1	40.2, CH_2_	1.55, m ^c^	41.9, CH_2_	1.63, d (13.2)	40.4, CH_2_	1.60, d (13.2)
		0.66, m		0.89, m		0.72, m
2	19.3, CH_2_	1.55, m ^c^	28.1, CH_2_	1.75, t (11.4) ^c^	19.4, CH_2_	1.54, m
		1.33, m ^c^		1.40, m		1.33, m ^c^
3	42.7, CH_2_	1.33, m ^c^	40.1, CH_2_	1.51, m ^c^	41.7, CH_2_	1.73, d (13.2)
		1.10, m		0.48, t (9.6)		0.76, m ^c^
4	33.8, C		33.7, C		38.1, C	
5	57.0, CH	0.68, m	56.5, CH	0.55, d (12.6)	57.2, CH	0.76, m ^c^
6	18.6, CH_2_	1.47, m	19.3, CH_2_	1.28, m ^c^	19.0, CH_2_	1.47, d (13.8)
		1.35, m				1.33, m ^c^
7	41.3, CH_2_	1.84, d (12.6)	42.8, CH_2_	1.31, m ^c^	42.8, CH_2_	1.36, m
		0.74, m				1.16, dd (13.2, 9.6)
8	37.9, C		38.2, C		38.0, C	
9	58.1, CH	0.81, m	58.7, CH	0.95, m	59.4, CH	0.86, m
10	33.9, C		37.8, C		34.0, C	
11	28.1, CH_2_	1.96, d (11.4)	28.0, CH_2_	1.96, dd (11.6, 3.6)	27.9, CH_2_	2.01, m
		1.74, q (12.0)		1.75, t (11.4) ^c^		1.79, m
12	72.8, CH	4.57, dd (11.4, 4.2)	76.6, CH	4.24, dd (11.4, 3.6)	77.0, CH	3.74, dd (10.8, 3.6)
13	44.9, C		42.5, C		43.9, C	
14	57.9, CH	2.19, brs	47.8, CH	2.29, m ^c^	53.1, CH	1.17, d (12.6)
15	133.6, CH	6.11, dd (9.6, 1.8)	25.4, CH_2_	2.48, m	28.1, CH_2_	2.37, dd (12.6, 7.2)
				2.29, m ^c^		2.01, m
16	118.4, CH	6.57, dd (9.6, 2.4)	153.7, CH	7.07, m	69.4, CH	5.13, dd (8.7, 7.5)
17	126.3, C		140.1, C		133.0, C	
18	167.3, C		52.1, CH	4.27, s	149.7, CH	7.94, s
19	99.1, CH	7.02, s	176.6, C		172.0, C	
20	170.8, C		194.4, CH	9.71, s	33.9, CH_3_	0.88, s
21	33.8, CH_3_	0.85, s	33.8, CH_3_	0.81, s ^c^	22.0, CH_3_	0.81, s ^c^
22	21.9, CH_3_	0.80, s ^c^	22.0, CH_3_	0.77, s	16.9, CH_3_	0.81, s ^c^
23	16.6, CH_3_	0.80, s ^c^	17.3, CH_3_	0.81, s ^c^	18.2, CH_3_	0.91, s
24	19.7, CH_3_	1.01, s	17.6, CH_3_	0.93, s	16.9, CH_3_	1.40, s
25	12.5, CH_3_	1.26, s	16.8, CH_3_	1.13, s		

^a^ Data obtained in pyridine-*d*_5_. ^b^ Carbons correlating with the corresponding protons in the HSQC spectrum. ^c^ Overlapped with other signals.

**Table 4 marinedrugs-20-00604-t004:** Inhibitory activities of the isolated compounds **1**−**18** against cancer cell growth (IC_50_, μM) ^a^.

Compound	Cell Line		Compounds	Cell Line	
	HeLa	MCF-7		HeLa	MCF-7
**1**	53.4	27.3	**11**	>80.0	49.4
**2**	46.2	26.2	**12**	41.9	24.3
**3**	60.1	29.9	**13**	65.2	40.8
**4**	61.3	45.9	**14**	58.4	30.9
**5**	70.7	76.4	**15**	>80.0	51.3
**6**	59.3	33.8	**16**	46.3	20.0
**7**	>80.0	>80.0	**17**	46.9	36.5
**8**	>80.0	>80.0	**18**	>80.0	>80.0
**9**	>80.0	43.8			
**10**	45.7	27.7	staurosporine	0.18	0.13

^a^ Data are an average of at least two tests.

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
