# Peer review of "Scalarane Sesterterpenoids Isolated from the Marine Sponge Hyrtios erectus and their Cytotoxicity"

_marinedrugs, 2022, doi:10.3390/md20100604_

Round 1

Reviewer 1 Report

The manuscript entitled “Scalarane Sesterterpenoids Isolated from the Marine Sponge Hyrtios erectus and their Cytotoxicity against HeLa and MCF-7 Cancer Cell Lines” describes the discovery 18 new scalarane sesterterpenoids from marine sponge Hyrtios erectus. Eight compounds (1-8) were determined to be undescribed. Compounds 1-3, 10, 12, and 16 showed weak cytotoxicity towards HeLa and MCF-7 cancer cells (IC50 >=20.0 μΜ). Overall, the structure elucidation is logical. The paper can be published in Mar. Drugs after major revisions.

Here are some comments about the manuscript, which should be addressed by the authors.

1.    Abstract: γ-Hydroxybutenolide moiety is pretty common in marine sponge products. They are not unique. Also, based on the activity observed, it is not true that this motif is crucial for the cytotoxicity. At least, the most active compound 16 in this paper does not possess this motif. Besides, compounds no strong structure activity relationship was observed in this paper. So, authors should delete this sentence. Activity with IC50 values ranging from 24.3 to 29.9 μΜ is weak to moderate activity, not promising.

2.    Lines 76-77. For me, the typical 1H NMR signals for these compounds should be the signals for the scalarane skeleton (methyls and methylenes on the ring). Showing signals from δH 6.0 to 7.5 ppm and around 9.5 ppm for what? Authors should re-write this.

3.    Line 82. They are just simple analogs of a well-known class of compounds. Please stop using novel in the manuscript.

4.    Lines 135-136: absolute stereochemistry cannot be determined by ROESY. In fact, authors did not figure out the absolute configurations of the isolates. Authors should use ECD calculation or Mosher’s method to determine absolute stereochemistry (e.g. OHs at C-12 and C-19 in 6).

Author Response

(Reviewer 1)

Reviewer’s comment

Response

Abstract: γ-Hydroxybutenolide moiety is pretty common in marine sponge products. They are not unique. Also, based on the activity observed, it is not true that this motif is crucial for the cytotoxicity. At least, the most active compound 16 in this paper does not possess this motif. Besides, compounds no strong structure activity relationship was observed in this paper. So, authors should delete this sentence. Activity with IC50 values ranging from 24.3 to 29.9 μΜ is weak to moderate activity, not promising.

We have edited the entire manuscript as pointed out. In particular, the description of the level of cytotoxicity was corrected to 'weak'.

Lines 76-77. For me, the typical 1H NMR signals for these compounds should be the signals for the scalarane skeleton (methyls and methylenes on the ring). Showing signals from δH 6.0 to 7.5 ppm and around 9.5 ppm for what? Authors should re-write this.

Based on the 1H NMR of the n-hexane fraction, it was speculated that it may contain secondary metabolites. The type of compounds could not be specified at this stage. As the initial description in section 2.1 was misleading and that in section 3 was incorrect, both have been modified.

Line 82. They are just simple analogs of a well-known class of compounds. Please stop using novel in the manuscript.

The word ‘novel’ has been replaced with ‘new’ throughout the manuscript.

Lines 135-136: absolute stereochemistry cannot be determined by ROESY. In fact, authors did not figure out the absolute configurations of the isolates. Authors should use ECD calculation or Mosher’s method to determine absolute stereochemistry (e.g. OHs at C-12 and C-19 in 6).

The absolute stereochemistry of the scalarane scaffold was taken from the previous literature, and the relative stereochemistry at each functionalized carbon was revealed in this study. Since the initial description was misleading, we’ve added a description of the absolute stereochemistry of the tetracyclic carbon framework with references and corrected the description of the relative stereo-chemistry description.

Reviewer 2 Report

The MS titled ``Scalarane Sesterterpenoids Isolated from the Marine Sponge Hyrtios erectus and their Cytotoxicity against HeLa and MCF-7 Cancer Cell Lines`` can not be accepted for publication in the current form. The below issue should be carefully addressed.

Remove cell lines from the title

Don't use novel, use new in the abstract and text, because the core skeleton is already known. 

The absolute stereochemistry cannot be established by RoEsy, refer to it as relative stereochemistry. 

Write the full name of all abbreviations, when they are firstly appeared. 

Sponge family name should be added in the abstract and keywords

Line 18 in the abstract need revision, it is confusing.

Type of assay and control as well as the results should be added in the abstract

Remove the pronoun "we".

Introduction lacks details about the sponge and its reported metabolites and their bioactivity. 

For line 98 and 99, a reference should be added. 

The cosy and HMBC correlations of compound 1, as well as HSQC, should be discussed, and various spin system establishment. This compound should be fully discussed. 

In line 144, a reference should be added. 

In line 155, "ant" correct. 

Careful revision for English is needed. There are some typing and grammatical mistakes. 

In compound 3, the authors should discuss the CoSY and HMBC correlations that supported the absence of the C-16 methoxy group.

The absolute stereochemistry couldn't be assigned by RoESY and thus replace absolutely by the relative 

Additionally, there is a difference in the position of the carbonyl group in compounds 1 and 2 they have OH at C-20 and carbonyl at C-19, while 4 and 5 the carbonyl at C-20 and OH at C-19, this should be discussed.

The 2DNMR (CoSY, HMBC, and HSQC) data for compounds 4-8 as well as  their roles in structure assignment and establishment of the substitution should be discussed. 

The carbon multiplicity ( CH, CH2,.... ) should be added to the tables.

Comparing the activity of the tested compounds and control showed that they have weak and very weak activity, no moderate effect, check and correct.

To add more value to this work, their biosynthetic pathways should be included.

Date of sponge collection should be added.

The polarity of the solvent system corresponding to each fraction should be added.

Check the colors of the isolated compounds in the results and experimental sections they are different.

The conclusion is missing.

All references should be formatted correctly according to the journal style.

Author Response

(Reviewer 2)

Reviewer’s comment

Response

Remove cell lines from the title

The title has been modified as pointed out.

Don't use novel, use new in the abstract and text, because the core skeleton is already known. 

The word ‘novel’ has been replaced with ‘new’ throughout the manuscript.

The absolute stereochemistry cannot be established by RoEsy, refer to it as relative stereochemistry. 

The absolute stereochemistry of the scalarane scaffold was taken from the previous literature, and the relative stereochemistry at each functionalized carbon was revealed in this study. Since the initial description was misleading, we’ve added a description of the absolute stereochemistry of the tetracyclic carbon framework with references and corrected the description of the relative stereochemistry description.

Write the full name of all abbreviations, when they are firstly appeared. 

The modification has been made throughout the manuscript as suggested.

Sponge family name should be added in the abstract and keywords

The family name has been added in the abstract and keywords. 

Line 18 in the abstract need revision, it is confusing.

The abstract has been modified for clarity and objectivity. 

Type of assay and control as well as the results should be added in the abstract

The information on cytotoxicity measurement have been added in the abstract for clarity.

Remove the pronoun "we".

The use of the word ‘we’ has been excluded from the manuscript.

Introduction lacks details about the sponge and its reported metabolites and their bioactivity. 

The description of the metabolites from the Hyrtios sponges and their biological activity has been added in the introduction with references.

For line 98 and 99, a reference should be added.

The reference has been added.

The cosy and HMBC correlations of compound 1, as well as HSQC, should be discussed, and various spin system establishment. This compound should be fully discussed.

The modification has been made as suggested.

In line 144, a reference should be added. 

We carefully scrutinized the manuscript to cover as many references as necessary.

In line 155, "ant" correct. 

Typos including this have been corrected.

Careful revision for English is needed. There are some typing and grammatical mistakes. 

We carefully proofread the manuscript on our own, and then had the help of a professional English proofreading company (Cactus communications)

In compound 3, the authors should discuss the CoSY and HMBC correlations that supported the absence of the C-16 methoxy group.

An explanation of the structural differences between compounds 2 and 3 was added.

Additionally, there is a difference in the position of the carbonyl group in compounds 1 and 2 they have OH at C-20 and carbonyl at C-19, while 4 and 5 the carbonyl at C-20 and OH at C-19, this should be discussed.

The description of the key HMBC correlations that made it possible to discriminate these regioisomers was added.

The 2DNMR (CoSY, HMBC, and HSQC) data for compounds 4-8 as well as their roles in structure assignment and establishment of the substitution should be discussed. 

The description has been modified and supplemented.

The carbon multiplicity ( CH, CH2,.... ) should be added to the tables.

The tables have been modified as suggested

Comparing the activity of the tested compounds and control showed that they have weak and very weak activity, no moderate effect, check and correct.

We have edited the manuscript as pointed out. In particular, the description of the level of cytotoxicity was revised to 'weak'.

To add more value to this work, their biosynthetic pathways should be included.

The plausible biosynthetic pathway for isolated scalarane derivatives is provided in the discussion session

Date of sponge collection should be added.

More information has been added for sponge collection

The polarity of the solvent system corresponding to each fraction should be added.

The solvent condition for each fraction has been added.

Check the colors of the isolated compounds in the results and experimental sections they are different.

The descriptions of the compound's color have been corrected.          

The conclusion is missing.

The conclusions are combined in the discussion section. The ‘instruction for authors’ of Marine Drugs indicates that the conclusion section is not mandatory but can be added to the manuscript if the discussion is unusually long and complex. 

All references should be formatted correctly according to the journal style.

The reference section has been carefully edited. 

Reviewer 3 Report

Dear Editor,

The authors showed "calarane Sesterterpenoids Isolated from the Marine Sponge Hyrtios erectus and their Cytotoxicity against HeLa and MCF-7 Cancer Cell Lines" The study is interesting. However, several points in this paper should be reconsidered:

-The biological test has been used in this study is not enough to demonstrate the anti-cancer activity of compounds. MTS assay is a simple test to demonstrate the cytoxcicity of compounds. To demonstrate the anti-cancer activity of compounds some biological tests including flow cytometry for apoptosis detection, Real Time-PCR of specific genes that involve in apoptosis and anti-proliferative activity, and … are required.

-The discussion part can be improved by providing a more critical discussion of recent related literature. Discuss the shortcomings of previous work and the gaps and how this work intends to fill those gaps. Cytotoxicity and apoptosis assay in the discussion should be improved.

-It is suggested that the authors report the results of the cytotoxicity of compounds against the normal cell line.

-References should be updated. The new references should be added (in both the introduction and discussion).

Author Response

(Reviewer 3)

Reviewer’s comment

Response

The biological test has been used in this study is not enough to demonstrate the anti-cancer activity of compounds. MTS assay is a simple test to demonstrate the cytoxcicity of compounds. To demonstrate the anti-cancer activity of compounds some biological tests including flow cytometry for apoptosis detection, Real Time-PCR of specific genes that involve in apoptosis and anti-proliferative activity, and … are required.

It is suggested that the authors report the results of the cytotoxicity of compounds against the normal cell line.

Additional experiments such as the measurement of cytotoxicity against normal cell lines, cell migration/invasion assay, flow cytometry analysis, and apoptosis biomarker analysis were planned initially for the compounds with potent cytotoxicity against cancer cell lines. However, the cytotoxicity of isolated compounds was disappointing, which made us file the report at this stage without further experiments.

The discussion part can be improved by providing a more critical discussion of recent related literature. Discuss the shortcomings of previous work and the gaps and how this work intends to fill those gaps. Cytotoxicity and apoptosis assay in the discussion should be improved.

References should be updated. The new references should be added (in both the introduction and discussion).

We attempted to enhance the discussion session by proposing plausible biosynthetic pathways for isolated compounds. We also reviewed the literature again to reference the latest publications throughout the manuscript.              

Round 2

Reviewer 1 Report

Please provide Figure 2 with higher resolution. Also, I personally prefer elucidating HMBC correlations or even ROESY in 2D structures. Authors could try and see which is better.

Reviewer 2 Report

Remove the cell lines from the title as suggested previously.

The sponge genus and species name should be italicized throughout the whole MS.

Reviewer 3 Report

The authors have responded adequately to my previous comments. I have no more comments.